# Genetics of Host Protection against *Helicobacter pylori* Infections

**DOI:** 10.3390/ijms22063192

**Published:** 2021-03-21

**Authors:** Rosanna Capparelli, Domenico Iannelli

**Affiliations:** Department of Agriculture Sciences, University of Naples “Federico II”, via Università, 100-Portici, 80055 Naples, Italy

**Keywords:** *Helicobacter pylori*, resistance genes, metabolism

## Abstract

This narrative review discusses the genetics of protection against *Helicobacter pylori* (*Hp*) infection. After a brief overview of the importance of studying infectious disease genes, we provide a detailed account of the properties of *Hp*, with a view to those relevant for our topic. *Hp* displays a very high level of genetic diversity, detectable even between single colonies from the same patient. The high genetic diversity of *Hp* can be evaded by stratifying patients according to the infecting *Hp* strain. This approach enhances the power and replication of the study. Scanning for single nucleotide polymorphisms is generally not successful since genes rarely work alone. We suggest selecting genes to study from among members of the same family, which are therefore inclined to cooperate. Further, extending the analysis to the metabolism would significantly enhance the power of the study. This combined approach displays the protective role of *MyD88, TIRAP,* and *IL1RL1* against *Hp* infection. Finally, several studies in humans have demonstrated that the blood T cell levels are under the genetic control of the CD39^+^ T regulatory cells (T_REGS_).

## 1. Introduction

Evidence is clear that protection against pathogens is in part genetic. This evidence is provided by human genetic variants conferring resistance to different pathogens: the sickle-cell trait to malaria caused by *Plasmodium falciparum* [1], the absence of the Duffy blood group to malaria caused by *Plasmodium vivax* [2], and a variant of the C-C chemokine Receptor type 5 (CCR5) chemokine to human immunodeficiency virus (HIV) infection [3]. However, the impact of infectious diseases is still present, though considerably reduced by modern medicine. The persistence of diseases such as malaria, tuberculosis, the COVID-19 pandemic, and the widespread bacterial antibiotic resistance remind us of the importance of gaining a better understanding of infectious disease genetics.

In this review, we first illustrate the strategy used by *Helicobacter pylori* (*Hp*) to create a long-term relationship with the human host. We then describe the approaches more frequently used to identify the genes conferring resistance to pathogens. Finally, we discuss how the knowledge of host-*Hp* interaction might help the reader to find the best way to approach further study.

## 2. *Helicobacter pylori*–Human Host Interaction

*Hp*, a Gram-negative bacterium, colonizes the gastric mucosa of about 50% of the world population [4]. When present, *Hp* becomes the predominant component of the stomach microbiota [5]. This result suggests that the altered microbiota of the stomach might potentially influence the microbiome and immune system of the host. *Hp* infection mostly occurs during childhood by vertical transmission from mother to child or by horizontal transmission from infected siblings [5]. According to evolutionary theory, when transmitted vertically, pathogens evolve toward reduced virulence [6]. Consistent with this theory, most *Hp* infections remain inactive for several decades [7]. However, *Hp* can also be transmitted horizontally. In this case, infection with multiple strains disrupts the reduced virulence gained by vertical selection, and the bacterium can return virulent [6].

During its long coevolution with humans [7], *Hp* gained a very high level of genetic diversity through recombination with other strains during mixed infections [8]. At present, genetic differences are observed even between single colonies from the same patient. *Hp* populations also migrate to specific areas of the stomach [9]. Adaptation to individual niches and genetic recombination produce a well-structured protection against host immunity and antimicrobials (if one strain succumbs, very likely others will survive). *Hp* uses genetic recombination to alter the expression of the host surface antigens and thus escape recognition by the host immune system [9].

During its long coevolution with the host, *Hp*, in addition to genetic recombination, developed several more obstacles to host immunity. Vacuolating cytotoxin A (VacA) and Cytotoxin-associated gene A (CagA), - two toxins causing adenocarcinoma and mucose-associated lymphoid tissue (MALT) lymphoma - induce apoptosis [10], whereas urease controls gastric acidity [11]. Then, lipopolysaccharide (LPS) escapes recognition by phase variation, a process that helps bacteria to exhibit different LPS epitopes. *Hp* expresses the human blood group O-antigen. This molecular mimicry trick enables *Hp* to evade the Toll-like receptors (TLRs) that recognize the O-antigen as self. In addition, *Hp* alters the net charge of the lipid A portion of LPS, making lipid A highly resistant to the cationic antimicrobial peptides (CAMPs) [12]. Flagellin (a protein of flagella) escapes recognition by Toll-like receptor 5 (TLR5) by expressing the less-inflammatory variant FlaA and catalase neutralizes the release of oxygen radicals from macrophages. This multilayered strategy enables *Hp* to efficiently evade host immunity and induce a chronic inflammation, compatible with long-term colonization of the host, but not apt to clear infection. 

*Hp* displays several more properties. Grown in the presence of low iron or high salt concentrations, the bacterium rapidly selects the carcinogenic variant FuR88H [13], which is associated with several non-gastric diseases [14]. Furthermore, *Hp* displays conflicting properties: it is the main risk factor for gastric carcinoma and gastric MALT lymphoma [7], but protects against esophageal adenocarcinoma, Barrett’s esophagus, and gastroesophageal reflux [15]. The conflicting roles of *Hp* in human diseases demand a clear understanding of its complex interactions with the host and the environment. A deeper knowledge of host–pathogen interactions may also help to decide with confidence whether humans are better off with or without *Hp* in their stomach. 

## 3. Why Study Infectious Disease Genes?

Detection of this class of genes helps in the preparation of new drugs and vaccines. Drug discovery against HIV was guided by studies showing that a deletion in the gene coding for the HIV coreceptor CCR5 reduces the risk of HIV infection [16]. The vaccine against malaria caused by *Plasmodium vivax* followed the evidence that absence of the Duffy blood group confers resistance against this pathogen [2]. Infectious disease genes also explain the contribution of pathogens to maintaining the genetic diversity of our genome. Human ABO blood groups and major histocompatibility complex (MHC) polymorphisms are maintained in the population because they protect against infectious diseases. The high frequency of cystic fibrosis in the population follows the advantage of heterozygotes for mutations in the chloride channel gene (*CFTR*) of being resistant to typhoid infection [17]. Evolution uses different mechanisms to maintain polymorphisms in the population: heterozygote advantage, frequency-dependent selection, and fluctuation in the selection pressure caused by its presence in the population of different strains of the same pathogen recognizing different host genotypes.

## 4. Detection of Infectious Disease Genes: Candidate Gene Studies

The genetics of resistance to pathogens has its roots in the thoughtful intuition of Haldane [18] and the experimental demonstrations provided by Allison in 1954, which established that, in humans, the gene causing sickle hemoglobin is associated with resistance to malaria caused by *Plasmodium falciparum*. Later studies demonstrated that the LTA4H gene (*LTA4H*) is associated with pulmonary tuberculosis, *PARK2* and *PACRG* with leprosy, and a mutant form of *CCR5* with reduced HIV-1 transmission [16]. Independent studies carried out in twin pairs demonstrated that the concordance rate of tuberculosis and *Hp* infection [19] was higher in monozygotic twin pairs than in dizygotic twin pairs. 

Out of the many outstanding candidate gene studies, we mention two that illustrate the following concepts: first, that the same gene can protect against multiple pathogens; and second, that landmark studies may also originate from the detailed analysis of a limited number of patients, rather than the survey of large cohorts.

A case-control study of patients from the U.K., Vietnam, and several African countries with invasive pneumococcal disease (IPD), bacteremia, malaria, or tuberculosis showed that patients heterozygous for the variant S180L of the protein Mal encoded by *TIRAP* are protected against the four diseases in all the study populations (P: 9.9 × 10^−8^) [20]. Following stimulation of *TLR2* and *TLR4*, the protein Mal triggered the activation of Nuclear Factor kappa-light-chain-enhancer of activated B cells (NF-kB) and the pro-inflammatory response [20]. In vitro studies demonstrated that the variant S180L curbs *NF-kB4* activation through the wild form of the Mal protein. Thus, heterozygosis at *S180L* protects against multiple diseases by providing a reduced immune response, proving that inflammation functions best when properly balanced. 

The signal transducer and activator of transcription1 (STAT1) controls the downstream type 1 interferon and several cytokine receptors expressed in many cell types. Loss-of-function mutations inhibiting the *STAT1* function cause susceptibility to viruses by inhibiting Interferon-α/β (IFN-α/β) and to mycobacterial diseases by inhibiting IFN-g [21]. In contrast, gain-of-function mutations in the same gene cause chronic mucocutaneous candidiasis by hampering the STAT1-dependent repressors of Interleukin-17 (IL-17)-producing T cells [21]. These studies illustrate the complexity of the in vivo relationship between host genes and pathogen, in particular, how mutations in the same gene can lead to different diseases by participating in multiple interactions, all causing different adverse consequences to the host. 

Unfortunately, the success and apparent simplicity of candidate gene studies also yielded a plethora of non-reproducible results.

## 5. Detection of Infectious Disease Genes: Genome-Wide Association Studies (GWAS) 

Genome-wide association studies (GWAS) offer the opportunity to test millions of single nucleotide polymorphisms (SNPs). However, the method also has serious limitations. Thousands of cases and controls are required to reach the requested statistical significance level (*p* < 5 × 10^−7^), a number too high to reach even in countries where infectious diseases are recurrent. In addition, GWAS can explain only 15–20% of the hereditability measured using twin studies [22]. Accordingly, very few infectious disease studies have been carried out using GWAS. The best GWAS in infectious diseases are those on leprosy, which identified five genes tightly associated with this disease [23].

## 6. Presence in the Population of Different Strains of the Same Pathogen

A review of highly reproducible infectious disease studies included tuberculosis, malaria, and leprosy [24]. This realization has been attributed to the low genetic variability of these pathogens, in particular of *Mycobacterium leprae* [25]. This conclusion suggested that the difficulties in replicating GWAS case-control studies and their low hereditability, at least in part, might reflect the presence in the population of multiple strains of the same pathogen [24]. Preliminary results confirmed that stratification of patients according to the infecting pathogen strain enhances both the power and replication of the study [26].

## 7. Infectious Disease Genes Controlled at the Transcriptional Level

Tumor necrosis factor-α (TNF-α) is involved in the pathogenesis of several diseases, including cerebral malaria, characterized by high levels of this cytokine [27]. TNF-α has two allelic forms located in the promoter at −311: *TNF1* and *TNF2*. The latter allele is associated with higher levels of TNF-α transcription than the former (*TNF1*). A case-control study of malaria in Gambia showed that *TNF2* homozygous patients are significantly more numerous among cases of cerebral malaria [27]. In Gambia, the *TNF2* allele reaches a frequency of 0.6, despite its association with cerebral malaria. This finding suggested that *TNF2* is maintained in the population because heterozygotes possess levels of TNF-α conferring optimal protection against diseases other than cerebral malaria. 

More recently, flow cytometry analysis of cell surface protein expression levels in 669 twin pairs demonstrated that the quantitative expression of several regulatory T cell (T_REGS_) proteins is under genetic control. One of the most hereditable traits is the CD39 protein expressed by CD39^+^ CD4 T_REGS_ [28]. Individuals homozygous for the SNP rs096317A expressed high levels of the CD39 protein, heterozygous individuals expressed intermediate levels, and those homozygous for the SNP rs0966317G did not express this protein at all [28]. The same study described multiple polymorphisms at several other loci of T_REGS_ cells that control surface protein expression. 

T_REGS_ cells also play a role in *Hp* infection [29]. The gastric mucosa inflammation caused by *Hp* infection is in part regulated by T_REGS_. CD4^+^/CD5^+^ T_REGS_ can suppress cytokine production of other T cells. The role of T_REGS_ in *Hp*-induced gastritis was studied in mice. Athymic mice were reconstituted with lymph node cells depleted of CD25^+^ cells or with CD25^+^ lymph node cells. Three weeks later, mice were infected with *Hp*. At six weeks from infection, the mice reconstituted with lymph node cells depleted of CD25^+^ cells developed a form of gastritis more severe than that of mice reconstituted with CD25^+^ lymph node cells. The experiment demonstrated that T_REGS_ CD25^+^ cells curb *Hp*-induced gastric mucosa inflammation [30]. 

Gene expression analysis via microarrays was also used to investigate how pathogens modulate the host’s gene expression [31], detect candidate genes conferring resistance to pathogens [18], identify the infecting pathogen [31], and explain why some patients infected with the hepatitis C virus do not respond to interferon therapy [17].

## 8. *Hp* Modulates Gene Expression through Epigenetics and Co-Infection 

Epigenetics describes reversible mechanisms that regulate gene expression without altering the DNA sequence [32]. Methyltransferases (MTs) are molecules that transfer DNA methyl groups from methionine to adenine or cytosine residues. MTs control the expression of a large number of bacteria, including *Hp* [32]. Almost every *Hp* strain has its unique set of MTs. Transcriptome analysis of two *Hp* strains (J99 and BCM-300) and their respective MTs mutants showed that inactivation of MTs leads to changes in the expression of 225 genes in strain J99 and 29 genes in strain BCM-300, altering bacterial adherence to host cells, natural competence for DNA uptake, and bacterial cell shape (Table 1) [32].

In patients infected with CagA+ strains of *Hp*, the methylation level of several tumor-suppressor genes was up to 300-fold higher than in non-infected individuals [33]. Silencing of tumor-suppressor genes by methylation sensibly increases the risk of gastric cancer [33]. To study how *Hp* infection influences methylation, Mongolian gerbils (*Meriones unguiculatus*) were infected with *Hp*. At 50 weeks from infection, the animals displayed levels of methylation up to 200-fold higher than controls [33]. Cyclosporine, which inhibits inflammation but not bacterial replication, prevented methylation. This result demonstrated that gene methylation is induced by *Hp*-infection-induced inflammation [34].

*Hp* and Epstein–Barr virus (*EBV*) share the property of inducing chronic inflammation in the host, which favors the development of cancer. Gastric epithelial cells infected with EBV, upon in vitro coinfection with *Hp*, display enhanced bacterial proliferation and oncogenic activity, both mediated by the bacterial protein CagA. *Hp*-*EBV* coinfection induces transcription of MTs, which silence tumor suppressor genes, causing altered cell cycle, apoptosis, and DNA repair genes [35]. In conclusion, epigenetics and coinfection are major areas to explore to define the role of *Hp* in the context of extragastric diseases, including cancer. 

## 9. Resistance to Pathogen May Be Ephemeral

This topic is rarely mentioned. To describe it, we refer to an iconic experiment known as “one of the greatest natural experiments in evolution” [36]. Rabbits, introduced in Australia by European settlers, caused serious economic and ecological damage. To control the rabbit population, in 1950, the myxoma virus was released in Australia, and in 1952, it was introduced in France, reaching the United Kingdom in 1953. In all three countries, a rapid decrease in rabbit mortality was observed along with an increase in rabbit resistance to the virus [36]. When resistance reduces the replication of the pathogen in the host rather than inhibiting infection, selection may evolve into an increase in pathogen virulence [37]. In line with this theory, the decline in virulence that followed the virus release, was replaced decades later by a highly virulent myxoma strain [38]. This classic experiment reminds us that pathogens can become more virulent in response to increased resistance of the host, unless genetic selection or vaccination completely inhibits transmission [39,40]. This is a gentle reminder to the people responsible for the ongoing COVID-19 vaccination plans.

## 10. Potential Role of *Hp* against Inflammatory and Autoimmune Diseases

To establish chronic infection of the host, bacteria first need to modulate the immune system of the host. The pathogen-associated molecular patterns (PAMPs) are molecules common to a class of bacteria and recognized by pattern recognition receptors (PRRs) such as Toll-like receptors. PRRS detect bacterial PAMPs and alert the innate immune response. In addition to PAMPs, bacteria have also immunoregulatory molecules that prevent bacterial clearance and enable chronic infection. *Hp* is particularly well-structured to establish chronic infection that, in the majority of cases, remains asymptomatic. Further, *Hp* protects the host against autoimmune diseases, asthma, and esophageal adenocarcinoma [41]. Chronic colonization and protection of the host against several diseases suggest that *Hp* might promote immune tolerance. This conclusion is validated by the evidence that *Hp* induces the production of IL-10, a cytokine with anti-inflammatory activity that promotes immune tolerance and enables colonization of gastric mucosa [42]. Transforming growth factor beta (TGF-β) controls inflammation induced by *Hp* and homeostasis through CD4^+^, CD25^+^ regulatory (T_REG_ cells). TGF-β secreted by CD4^+^ T_REG_ cells modulates cytokine production and the T cell immune response in lepromatous disease [43], Foxp3 gene expression, and T_REG_ production [44]. Host colonization, tolerance induction, and induction of immunoregulatory response require the role of the macrophage peroxisome proliferator-activated receptor gamma (PPARγ), an anti-inflammatory transcription factor [45]. These results suggest that *Hp* might represent a suitable system to identify the regulatory mechanisms controlling the host immune response [46].

Soon after infection, macrophages and dendritic cells undergo a drastic gene expression reprogramming, where interacting genes all express the same expression pattern (all up- or all downregulated). The loss of a single gene interacting with immunity and metabolism compromises the whole system. In particular, suppression or inactivation of PPARγ results in stronger inflammatory responses, while activation or enhanced expression of PPARγ leads to a more balanced response, maintained by activation of immunoregulatory pathways that control key metabolic events and limit the upregulation of inflammation genes [42]. 

In vitro cocultures of a wild type or of PPARγ-deficient bone-marrow-derived macrophages with live *Hp* identified several potential new immunoregulatory genes. One of them (Plexin domain-containing 2; *Plxdc2*) was confirmed to play an immunoregulatory role in the *Hp* infection, in a mouse model of inflammatory bowel disease, and potentially in other inflammatory and autoimmune diseases [46].

## 11. *Hp* and Metabolic Diseases 

Upregulation of *TORC1*, high levels of branched chain amino acids (BCAAs), inflammation, and mitochondrial dysfunction characterize *Hp* infection [47,48,49,50]. The same traits also characterize type 2 diabetes (T2D), obesity (OB), Alzheimer’s disease (AD), and cardiometabolic disease (CMD) [51,52,53]. These results stimulated further work to determine whether *Hp* has a role in these diseases [14]. The use of a conventional epidemiological study was excluded since it would have required a very large number of *Hp*-infected patients and as many controls. In addition, known and unknown confounding factors—in particular, the presence of multiple *Hp* strains in the same patient, a frequent event with *Hp* infections—would make the replication of results very difficult. An in vitro model of *Hp* infection was chosen. The human gastric carcinoma cell line MKN-28 was incubated for 2 h with *Hp* culture filtrate (*Hpcf*). The cells were then analyzed using nuclear magnetic resonance (NMR) and polymerase chain reaction (PCR) array technology. In the absence of inflammation, *mTORC1* is under the control of *C-MYC*; while in the presence of inflammation, it is instead under the control of *HIF1α* [54]. Upregulation of *HIF1α* and *mTORC1* (Table 2) indicates that MKN-28 cells, following incubation with *Hpcf*, display the inflammatory phenotype. This conclusion is confirmed by the production of TNF-α and Il-6 (Table 2). Mitochondrial dysfunction is documented by upregulation of the antioxidant superoxide dismutase *SOD2* (Table 2), as well as the high levels of amino acids, in particular of BCAAs (Figure 1A). High levels of BCAAs are a trait common to all the four diseases under investigation. These data allowed us to conclude that BCAAs are associated with the four diseases, but are insufficient to attribute a causal role to BCAAs. Despite the clear evidence that high levels of BCAAs anticipate T2D for many years, it is not yet known whether BCAAs cause insulin resistance or T2D [55]. Wisely, at present, we classify BCAAs as biomarkers of the four diseases.

Following incubation with *Hpcf*, MKN-28 cells show increased concentration of BCAAs, while the extracellular medium shows reduced concentration of BCAAs (Figure 1A,B). Since both *Hp* and MKN-28 cells are auxotrophic for amino acids, it can be deduced that the high levels of BCAAs detected in MKN-28 cells incubated with *Hpcf* derive from depletion of the culture medium. Seemingly, it may be difficult to assume that both *Hp* and humans have lost the genes coding for the synthesis of essential amino acids. However, upon examination, the loss of genes makes sense. Humans obtain essential amino acids from their diet and *Hp* finds them in its niche (the gastric mucosa). In this context, the corresponding genes are no longer adaptive. Then, either the genes are lost or undergo mutations and return adaptive, assuming a novel function. In short, gene loss is often a means to update the genome [56]; if the environment changes, genes must also change.

## 12. Conclusions

Several factors create challenges in identifying the genes that protect the host from infection with *Hp*. *Hp* displays a high level of genetic recombination. Genetic differences are observed even between single colonies from the same patient [8], making identification of the genes that protect the host from infection with this pathogen difficult. This obstacle can be bypassed by stratifying patients according to the infecting strain of the pathogen, thus enhancing the power and replication of the study [26]. Second, scanning for single nucleotide polymorphisms (SNPs) is generally not successful since genes rarely work alone. This problem can be overcome by selecting genes that are members of the same family [57] and therefore predisposed to cooperate. The study can be further improved by analyzing both genes and metabolites. The approach combining genes and metabolites was exploited in a study that included *MyD88, TIRAP,* and *IL1RL1*, members of the same pathway, with the first two being physically associated [58]. Acting in concert, these genes identified gene combinations protecting against *Hp* infection (OR: 0.10; P: 2.8 × 10^−17^), while nuclear magnetic resonance (NMR) detected host pathways specifically deregulated by *Hp* [59]. NMR distinguished *Hp*-infected patients heterozygous at the *IL1RL1* locus (AC) from those homozygous (AA and CC) on the basis of their metabolic differences. Further, the probability calculation indicated that the odds of the above genotype distribution being due to chance was 1.8 × 10^−12^. This result shows the under-appreciated opportunity offered by metabolomics to reach definitive conclusions when enrolling a small number of patients [58]. The selective power of metabolomics has been confirmed by an independent study [14].

## Figures and Tables

**Figure 1 ijms-22-03192-f001:**
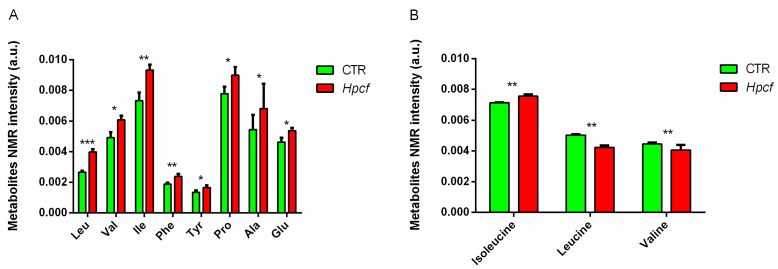
(**A**) Intracellular amino acid concentration differentiation (leucine, valine, isoleucine, phenylalanine, tyrosine, proline and alanine) detected in MKN-28 cells incubated (red columns) or not incubated (green columns) with *Hpcf*. (**B**) Extracellular branched chain amino acids (BCAAs) (leucine, isoleucine, and valine) concentration differences detected in culture medium of MKN-28 cells incubated (red columns) or not incubated (green columns) with *Hpcf*. Some *Hp* strains synthesize isoleucine, including ours, which explains the upregulation of isoleucine. The X-axis lists single amino acids, and the Y-axis reports the differences in individual amino acids scaled to the total NMR spectral area. Intensity of amino acids is expressed in arbitrary units and represented as means ± SD (* *p* < 0.5; ** *p* < 0.01; *** *p* < 0.001) calculated from two experiments, each carried out in quadruplicate (Reprinted with permission from ref. [14]. Copyright 2021, Domenico Iannelli).

**Table 1 ijms-22-03192-t001:** Genes involved in the epigenetic control of expression levels in *Hp* patients.

Biological Function	Gene ID	Gene Name
Signal transduction	*APC*	Adenomatous Polyposis Coli (APC) Regulator of WNT Signaling Pathway
	*RASSF1A*	Ras Association Domain Family Member 1
Cell cycle regulation	*CDH1*	Cadherin 1
	*CHFR*	Checkpoint with Forkhead and Ring Finger Domains
	*P14/ARF*	Cyclin-Dependent Kinase Inhibitor 2A
	*P15/INK4B*	Cyclin-Dependent Kinase Inhibitor 2B
	*P16/INK4A*	Cyclin-Dependent Kinase Inhibitor 2A
Inflammatory response	*COX-2*	Mitochondrially Encoded Cytochrome C Oxidase II
Apoptosis	*DAP-K*	Death-Associated Protein Kinase 1
DNA repair	*GSTP1*	Glutathione S-Transferase Pi 1
	*hMLH1*	MutL Homolog 1
	*MGMT*	O-6-Methylguanine-DNA Methyltransferase
Growth factor	*HPP1*	Hyperpigmentation, Progressive, 1
Transcription factor	*RUNX3*	RUNX Family Transcription Factor 3
Angiogenesis	*THBS1*	Thrombospondin 1
	*TIMP3*	TIMP Metallopeptidase Inhibitor 3

**Table 2 ijms-22-03192-t002:** Genes of mammalian Target of Rapamycin (mTOR) signaling, inflammatory, and oxidative stress pathways detected by polymerase chain reaction (PCR) array technology and differently expressed in MKN-28 cells incubated with *Hpcf* for 1 or 2 h. Variation of gene expression levels is reported as fold regulation. Values > |2| are considered statistically significant (Reprinted with permission from ref. [14]. Copyright 2021, Domenico Iannelli)).

Pathway Name	Gene ID	Gene Name	Fold Regulation 1 h	Fold Regulation 2 h
mTOR signaling pathway	*RPTOR*	Regulatory associated protein of mTOR complex 1	−1.42	286.04
	*MLST8*	mTOR associated protein, LST8 homolog (S. cerevisiae)	−1.42	398.95
	*AKT1*	V-akt murine thymoma viral oncogene homolog 1	−1.42	50.13
	*AKT2*	V-akt murine thymoma viral oncogene homolog 2	−1.42	504.97
	*INSR*	Insulin receptor	−1.42	257.79
	*IRS1*	Insulin receptor substrate 1	−1.42	278.22
	*PLD1*	Phospholipase D1, phosphatidylcholine-specific	−6.31	130.70
	*RPS6KA2*	Ribosomal protein S6 kinase, 90kDa, polypeptide 2	−1.24	3.37
	*PDPK1*	3-phosphoinositide dependent protein kinase-1	−1.53	28.25
	*PIK3CB*	Phosphoinositide-3-kinase, catalytic, beta polypeptide	−1.42	16.34
	*PIK3CD*	Phosphoinositide-3-kinase, catalytic, delta polypeptide	3.37	184.83
	*PIK3CG*	Phosphoinositide-3-kinase, catalytic, gamma polypeptide	−1.42	215.28
	*CHUK*	Conserved helix-loop-helix ubiquitous kinase	−4.08	181.03
	*EIF4E*	Eukaryotic translation initiation factor 4E	−1.42	922.92
	*HIFIA*	Hypoxia inducible factor 1, alpha subunit	192.93	955.47
Inflammatory pathway	*CXCL8*	Interleukin 8	−3.29	2.96
	*IL-6*	Interleukin 6	14.45	114.56
	*TLR2*	Toll-like receptor 2	58	72.18
	*TLR9*	Toll-like receptor 9	3.29	134.55
	*TNF*	Tumor necrosis factor	12.9	154.26
Oxidative stress pathway	*ATOX1*	ATX1 antioxidant protein 1 homolog (yeast)	3.57	37.69
	*GPX2*	Glutathione peroxidase 2 (gastrointestinal)	3.57	37.69
	*GPX4*	Glutathione peroxidase 4 (gastrointestinal)	3.57	37.69
	*GSS*	Glutathione synthetase	3.57	9.54
	*NOX5*	NADPH oxidase, EF-hand calcium binding domain 5	3.57	7.54
	*SOD1*	Superoxide dismutase 1, soluble	−28.68	−9.67
	*SOD2*	Superoxide dismutase 2, mitochondrial	3.96	4.04

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
