# Peer review of "Genetics of Host Protection against Helicobacter pylori Infections"

_ijms, 2021, doi:10.3390/ijms22063192_

Round 1
Reviewer 1 Report
The paper is interesting and well written.
I have only a suggestion:
- can the authors include a table to summarize the epigenetic control of gene expression mediated by Hp?
Author Response
The paper is interesting and well written.
I have only a suggestion:
- can the authors include a table to summarize the epigenetic control of gene expression mediated by Hp?
Please see Table 1.
Reviewer 2 Report
I believe that the manuscript presented by the Authors on the genetics of protection against infections caused by pathogenic microorganisms (especially H. pylori) is interesting and worth publishing.
Below I present a few minor adjustments, the inclusion of which will improve the quality of the article:
- I propose to change the title to “Genetics of host protection against Helicobacter pylori infections” (the phrase "resistance" is very strongly related to the insensitivity of microorganisms to antibiotics or other substances with antimicrobial activity and may be misinterpreted by some readers) [line 2]
- This term – “resistance” – should also be corrected in other part of this article, e.g. line 7 or 23
- “This review discusses the genetics of …” -> This narrative review discusses the genetics of … [line 7]
- “Finally, TREGS cells have a role in Hp infection.” -> what role? Please indicate [line 17]
- “next, the approaches more frequently used to identify the genes …” -> please add a verb after “next,” [line 33]
- Line 47: I think it is worth adding two or three sentences about the phase variation and its significance in the process of pylori virulence + please add appropriate references here
- “human tuberculosis” -> please delete, because this association is not that well discovered/ proved, compared to other diseased mentioned [line 71]
- “Plasmodium falciparum” -> should be Plasmodium falciparum (written using italics) [line 95]
- “Presence in the population of different strains of the same pathogen.” -> Presence in the population of different strains of the same pathogen (without a dot) [line 134]
- This conclusion is validated by the evidence that Hp induces the production of Il-10, a cytokine with anti-inflammatory activity that promotes immune tolerance and enables colonization of gastric mucosa [43].” -> please add some information about TGF-beta, a second important anti-inflammatory cytokine, and add appropriate reference here [lines 225-227]
- Please add references to Figure 1 and Table 1. because it is not known at present whether there are new results presented or some reference to already published ones (if the results are new, it is worth adding, for example, as a supplement a more detailed methodology of the presented research, bp. reference gene -> Table 1)
Author Response
I believe that the manuscript presented by the Authors on the genetics of protection against infections caused by pathogenic microorganisms (especially H. pylori) is interesting and worth publishing.
Below I present a few minor adjustments, the inclusion of which will improve the quality of the article:
The authors thank the reviewer for carefully reading our manuscript and for his suggestions that definitely will improve the quality of the article. The article has been reviewed accordingly with his suggestions. Below we point out - one by one - the line of each adjustment.
- I propose to change the title to “Genetics of host protection against Helicobacter pylori infections” (the phrase "resistance" is very strongly related to the insensitivity of microorganisms to antibiotics or other substances with antimicrobial activity and may be misinterpreted by some readers) [line 2]
Please see line 2.
- This term – “resistance” – should also be corrected in other part of this article, e.g. line 7 or 23
See lines 8 and 23.
- “This review discusses the genetics of …” -> This narrative review discusses the genetics of … [line 7]
See line 8.
- “Finally, TREGS cells have a role in Hp infection.” -> what role? Please indicate [line 17]
See line 17.
- “next, the approaches more frequently used to identify the genes …” -> please add a verb after “next,” [line 33]
See line 33.
- Line 47: I think it is worth adding two or three sentences about the phase variation and its significance in the process of pylori virulence + please add appropriate references here
See lines 59-63.
- “human tuberculosis” -> please delete, because this association is not that well discovered/ proved, compared to other diseased mentioned [line 71]
See line 73.
- “Plasmodium falciparum” -> should be Plasmodium falciparum (written using italics) [line 95]
See line 97.
- “Presence in the population of different strains of the same pathogen.” -> Presence in the population of different strains of the same pathogen (without a dot) [line 134]
See line 136.
- This conclusion is validated by the evidence that Hp induces the production of Il-10, a cytokine with anti-inflammatory activity that promotes immune tolerance and enables colonization of gastric mucosa [43].” -> please add some information about TGF-beta, a second important anti-inflammatory cytokine, and add appropriate reference here [lines 225-227]
See lines 234-237.
- Please add references to Figure 1 and Table 1. because it is not known at present whether there are new results presented or some reference to already published ones (if the results are new, it is worth adding, for example, as a supplement a more detailed methodology of the presented research, bp. reference gene -> Table 1)
See legend to Figure 1 and Table 2.